# Proteome-wide association study of prostate cancer risk across populations

Hua Zhong[1,2,17], Jingjing Zhu[2,3,17], Shuai Liu[1,2], Chong Wu [4], Liang Wang [5], Seamus P. Whelton[6], Catherine H. Marshall[7], Michael J. Blaha[8], Peter Durda[9], Xiuqing Guo [10], Craig W. Johnson [11], Henry J. Lin [10], Kent D. Taylor[10], Russell P. Tracy[9], Ronit I. Yarden[12], Ani W. Manichaikul [13], Stephen S. Rich [13], Jerome I. Rotter [10], Rajat Deo[14], Ruth F. Dubin [15], Peter Ganz [16] & Lang Wu [1,2] ✉

There is insufficient understanding of the molecular basis of prostate cancer (PCa) across different populations. We perform a large-scale proteome-wide association study (PWAS) to identify proteins with genetically regulated expression in plasma to be associated with PCa risk across populations. We develop genetic prediction models for expression of 1578, 1993, 1218, and 1390 proteins for African (n = 450), European (n = 758), Asian (n = 289), and Hispanic/Latino (n = 474) males, respectively, and evaluate associations of genetically regulated protein expression with PCa risk in 19,391 PCa cases and 61,608 controls of African population, 122,188 cases and 604,640 controls of European population, 10,809 cases and 95,790 controls of Asian population, and 3931 cases and 26,405 controls of Hispanic/Latino population. We identify three, four, 15, and 73 PCa-associated proteins in African, Hispanic/Latino, Asian, and European populations, respectively, and 83 in trans-population meta-analysis. There are both pan-population and population-specific associations. Our findings provide valuable insights into etiology of PCa.

Prostate cancer (PCa) is the most frequently diagnosed cancer among men in 118 countries/territories and the leading cause of cancer death in 52 countries[1]. There is a critical need to better understand its etiology for developing improved therapeutic and risk assessment strategies. Genetic factors have been demonstrated to play an important role in PCa etiology. It is well established that there is a significantly increased risk of PCa among men with a family history of the disease[2]. Genome-wide association studies (GWAS) have contributed significantly to our understanding of the heritability and familial risk of PCa[3,4]. Numerous genomic loci associated with PCa risk have been identified in GWAS[3,4], suggesting a multigenic model of prostate tumorigenesis. On the other hand, the underlying biological mechanisms for a majority of the GWAS-identified risk loci remain unclear. Given the constraints inherent in conventional single-marker-based GWAS[5], there has been a significant shift towards elucidating the functional implications underlying risk loci in the post-GWAS era. This emphasis aims to accurately delineate the exact biological mechanisms, particularly in post-translational processes through which the identified single nucleotide polymorphism (SNP)s and target genes exert their effects[6–11].

It is known that the proteome is commonly dysregulated during the development of diseases. Several proteins have been reported to be associated with PCa risk for their measured levels in blood, such as KLK11 (kallikrein related peptidase 11)[12], IL-6 (interleukin 6)[13], and EPCA (early PCa antigen)[14]. However, the findings are not always consistent[15,16]. Conventional epidemiological studies could confer several limitations, such as selection bias, potential confounding, and reverse causation, which may account for some of the inconsistent results observed. One alternative strategy is to evaluate the associations between genetically predicted protein levels and PCa risk. To identify proteins potentially relevant to PCa, we have comprehensively evaluated the relationship between genetically predicted protein

levels and PCa risk in European population[17,18] and identified a potential link between several protein biomarkers and PCa risk, including Laminin, IL-21, and HOXB13 (homeobox B13)[18,19].

Racial and ethnic background is a well-established risk factor for PCa[20,21]. PCa incidence rates vary significantly by regions, with the lowest rates observed in South Central Asia (6.4 age-standardized rate per 100,000), and markedly higher rates in Northern Europe (82.8 age-standardized rate per 100,000)[1]. Meanwhile, PCa mortality rates among men of African population are two to four times higher than those observed in men of other racial and ethnic groups[1,22]. Biological differences in PCa have also been observed across populations. For instance, the prostate-specific $Ca^{2+}$-dependent chloride ion channel protein[23], ANO7 (anoctamin 7), has been identified as a population-relevant protein-altering PCa-risk locus[24,25]. Two variants in the *ANO7* gene, rs74804606(p.Ile740Leu) and rs60985508 (p.Ser914*), have been identified to be associated with PCa in men of African ancestry[25]. In contrast, four European-specific PCa-risk variants in *ANO7*, rs77559646, rs2074840, p.Ala759Thr (rs76832527), and p.Glu226Lys/p.Glu226* (rs77482050), were largely excluded from analyses in African, East Asian, and Hispanic populations due to low frequency. Additionally, African-American men are known to exhibit higher expression of gene sets involved in the immune response, apoptosis, hypoxia, and reactive oxygen species production[26]. Despite these well-established differences, a comprehensive characterization of population-specific proteomic heterogeneity in PCa remains lacking, as most existing studies have focused primarily on men of European descent[19,27,28]. Although a previous study has examined population-specific exosomal proteins, its findings were limited by a very small sample size ($n = 12$ patients and nine controls) and a lack of adjustment for multiple comparisons[29].

In this work, we leverage a very large and comprehensive reference dataset of 7367 plasma proteins measured across four diverse populations − 450 African, 758 European, 289 Asian, and 474 Hispanic/Latino men − to establish population-specific genetic prediction models. We further analyze GWAS summary statistics for PCa risk, involving men in the African population (19,391 cases and 61,608 controls), European population (122,188 cases and 604,640 controls), Asian population (10,809 cases and 95,790 controls), and Hispanic/Latino population (3931 cases and 26,405 controls) to identify proteins whose genetically predicted levels are associated with PCa risk. Our study demonstrates the value of large-scale, diverse omics data for understanding the etiology of PCa to reduce its health disparities.

## Results

**Building population-specific protein genetic prediction models**
In this study, we first identified protein quantitative trait loci (pQTLs) for 450 African, 758 European, 289 Asian, and 474 Hispanic/Latino men residing in the USA in the Multi-Ethnic Study of Atherosclerosis (MESA) cohort. Among the 6999 proteins examined, we identified at least one *cis*-pQTL (within ±100 Kb of the transcription start site) associated at false discovery rate (FDR) < 0.05 and/or one *trans*-pQTL associated at $P < 5 \times 10^{-9}$ for 1609 proteins in the African population, 2058 in the European population, 1237 in the Asian population, and 1415 in the Hispanic/Latino population. We further established prediction models for 1578 proteins in the African population, 1993 in the European population, 1218 in the Asian population, and 1390 in the Hispanic/Latino population, each with an $R^2 \geq 0.01$, with 697 proteins overlapping across all genetic ancestries (Fig. 1a and Supplementary Data 1). Among the established prediction models, population-specific models were developed for 340 proteins in Africans, 515 in Europeans, 211 in Asians, and 156 in Hispanic/Latinos. The model performance $R^2$ varies across populations, ranging from 0.01 to 0.74 in Africans, 0.01 to 0.79 in Europeans, 0.01 to 0.77 in Asians, and 0.01 to 0.77 in Hispanic/Latinos, with Asians exhibiting the highest mean $R^2$ of 0.18 (Fig. 1b). Kruskal−Wallis test revealed a significant difference in $R^2$ values across the four groups ($\chi^2 = 98.39$, df = 3, $P < 2.20 \times 10^{-16}$). Post hoc Dunn's tests with Bonferroni correction confirmed that $R^2$ was significantly higher in Asians compared with Africans ($P = 2.57 \times 10^{-5}$), Europeans ($P = 7.81 \times 10^{-22}$), and Hispanic/Latinos ($P = 1.34 \times 10^{-5}$). It is worth noting that the Asian group was the smallest MESA population.

We further validated the European population prediction models using INTERVAL data in an external validation[30]. Due to differences

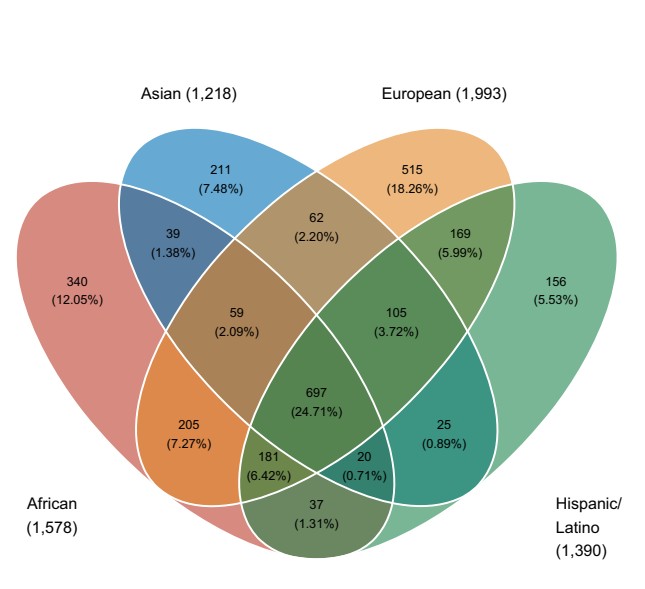
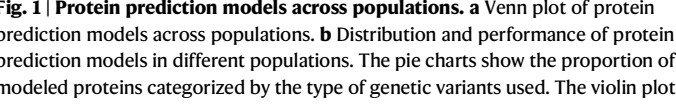

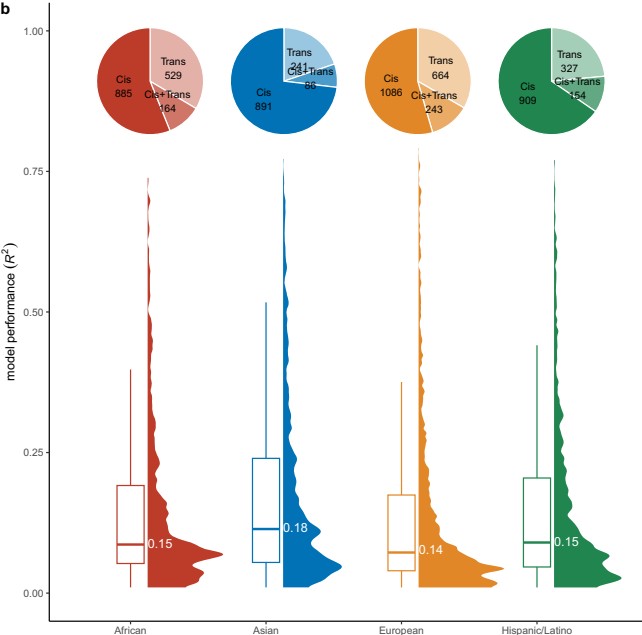

**Fig. 1 | Protein prediction models across populations. a** Venn plot of protein prediction models across populations. **b** Distribution and performance of protein prediction models in different populations. The pie charts show the proportion of modeled proteins categorized by the type of genetic variants used. The violin plot illustrates the distribution of $R^2$ values for protein models. The horizontal line within the box plot represents the mean value of $R^2$. The box boundaries define the interquartile range (IQR, 25th to 75th percentiles), and whiskers extend to the most extreme points within 1.5 × IQR. Source data are provided in the Source Data file.

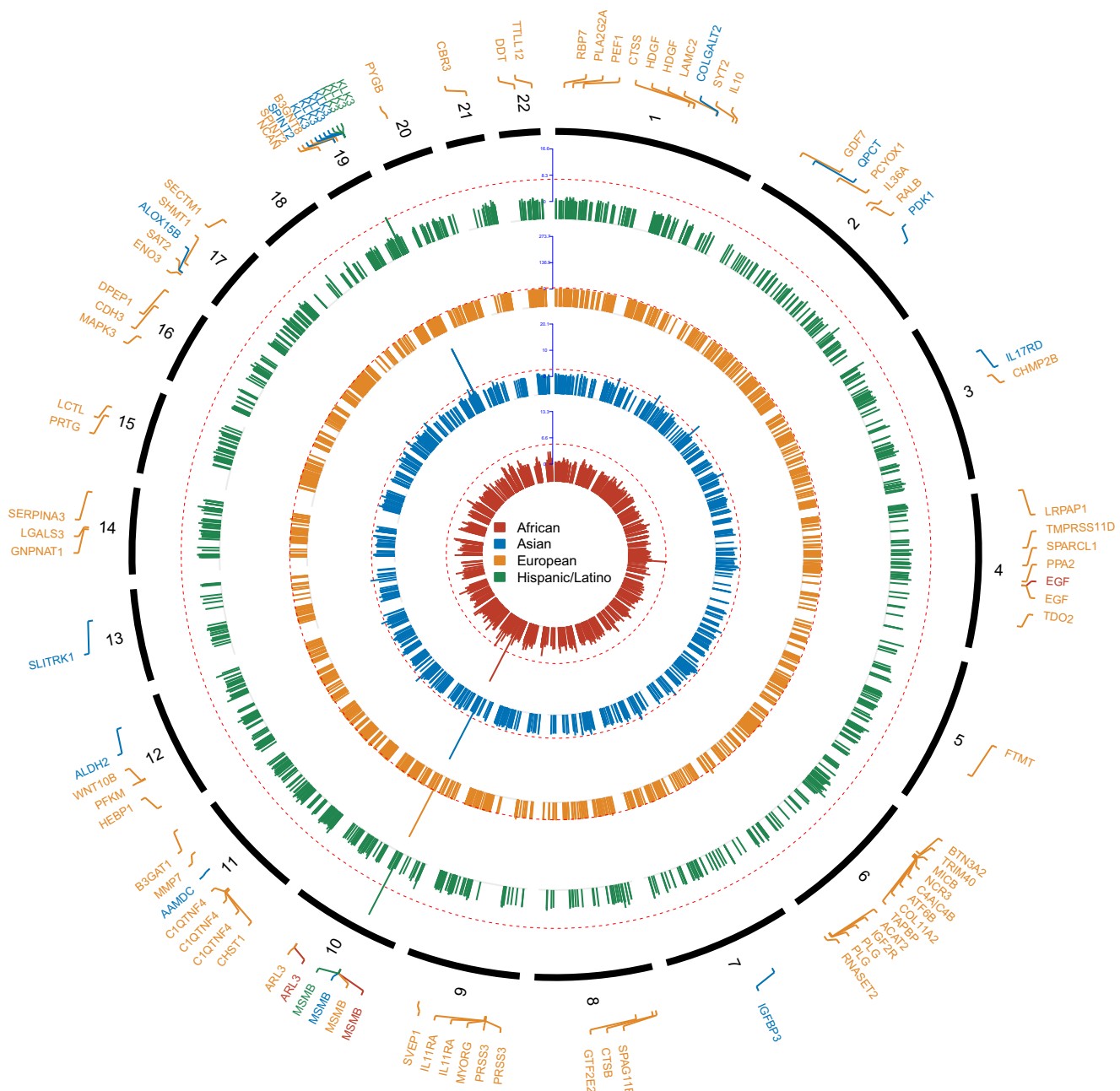

**Fig. 2 | Manhattan plot of identified proteins associated with prostate cancer (PCa) risk.** Each dot represents the *P*-value for the association between genetically predicted protein abundances and PCa risk, plotted by genomic position on the x-axis. Red dashed lines indicated the significant threshold at False Discovery Rate (FDR) < 0.05 for each population: African (19,391 PCa cases and 61,608 controls), European (122,188 cases and 604,640 controls), Asian (10,809 cases and 95,790 controls), and Hispanic/Latino (3931 cases and 26,405 controls). Source data are provided in the Source Data file.

between Somalogic's SomaScan platform versions, 844 proteins were available for analysis. Of these testable models, 584 (69.19%) exhibited a prediction performance of R² ≥ 0.01 in the external validation (Supplementary Fig. 1).

**Identification of proteins associated with PCa risk**
To identify proteins associated with PCa risk in each population of interest, we applied our established prediction models to the large-scale PCa GWAS summary statistics generated from 156,391 PCa cases and 666,248 controls across diverse populations. We identified significant associations (FDR < 0.05) for three proteins in the African population, namely, microseminoprotein-β (encoded by *MSMB*),

epidermal growth factor:extracellular domain (encoded by *EGF*), and ADP-ribosylation factor-like protein 3 (encoded by *ARL3*). In addition, 73 proteins in Europeans, 15 in Asians, and four in Hispanic/Latinos were significantly associated with PCa risk at FDR < 0.05 (Fig. 2 and Supplementary Data 2). Among them, microseminoprotein-β was detected across all four populations, and two proteins (ADP-ribosylation factor-like protein 3 and epidermal growth factor:extracellular domain) were identified in both African and European populations. Two unique proteins, prostate-specific antigen (PSA) and benign prostate-specific antigen (BPSA) encoded by *KLK3* (kallikrein related peptidase 3) were shared between Asian and Hispanic/Latino populations. A total of 60 proteins were unique to European population

(FDR < 0.05 in European and raw $p > 0.05$ in all other populations) and eight proteins tended to be Asian-specific (Supplementary Data 2).

A trans-population meta-analysis was conducted to identify proteins associated with PCa risk. A total of 92 proteins encoded by 83 genes were significantly associated with PCa risk at FDR < 0.05. Among them, 45 demonstrated inverse protein-PCa associations and 47 showed associations between higher predicted levels and increased PCa risk. Eighteen proteins not detected in population-specific analyses were identified in this meta-analysis (Supplementary Data 2). Based on a heterogeneity $p$-value (HetPval) of <0.05 or $I^2$ heterogeneity statistic (HetISq) > 75%, we found significant heterogeneity across populations for 10 unique proteins, including microseminoprotein-β, plasminogen, Charged multivesicular body protein 2b, Cathepsin S, Epidermal growth factor:Extracellular domain, tripartite motif-containing protein 40, Trypsin-3, Interleukin-36 alpha, PSA, Mth938 domain-containing protein, and UDP-GlcNAc:betaGal beta-1,3-N-acetylglucosaminyltransferase 8 (Supplementary Data 2).

Through population-specific proteome-wide association study (PWAS) and trans-population meta-analysis, we finally identified 96 protein-PCa risk associations, corresponding to 104 SOMAscan protein IDs. We then compared these 96 associated proteins with independent risk loci previously reported in GWAS[4,24]. Of these, 92 proteins were located within 500 kb of previously reported lead SNPs. The remaining four proteins (SLIT and NTRK-like protein 1, toll-like receptor 3, synaptotagmin-2, and complement factor H-related protein 1) were located more than 500 kb away from known loci (Supplementary Data 2).

## Two-stage constrained maximum likelihood (2ScML) robustness analysis

To assess the robustness of our results, we applied the 2ScML method under relaxed instrumental variable assumptions to 95 population-specific associations spanning 79 unique proteins with 86 SOMAscan IDs (three in Africans, 73 in Europeans, 15 in Asians, and four in Hispanic/Latinos). As shown in Supplementary Data 3, approximately 44.21% (42 of 95) of the associations remained significant at FDR < 0.05 with consistent effect directions. Except for PSA, detected in both Asian and Hispanic/Latino populations, the remaining 40 proteins were population-specific. Of these, 36 proteins tend to be specific to the Europeans and four were uniquely linked to PCa risk in the Asian population (aldehyde dehydrogenase, mitochondrial, arachidonate 15-lipoxygenase B, insulin-like growth factor-binding protein 3, and SLIT and NTRK-like protein 1).

## Tissue expression, functional enrichment, and network analysis of identified proteins

We evaluated the tissue-specific expression of 94 genes coding the 96 PWAS-identified proteins using RNA data from the Human Protein Atlas (HPA). Among these, three genes (*KLK3*, *MSMB*, and *ALOX15B*) exhibited prostate-enriched expression. These well-known PCa markers[31–33] validate our gene selection strategy. An additional 56 genes showed higher expression in other tissues but remained detectable in the prostate, many previously implicated in PCa-related pathways, including growth factor signaling (e.g., EGF[34], IGFBP3[35]), extracellular matrix remodeling and metastasis (e.g., PRSS3[36], MMP7[37]), and inflammatory regulation (e.g., SOCS3[38]).

Our Gene Ontology (GO) enrichment analysis of the 94 candidate genes revealed significant overrepresentation in relevant biological processes, including *insulin-like growth factor receptor signaling pathway* (GO:0048009, $P = 2.92 \times 10^{-5}$) and *extracellular matrix disassembly* (GO:0022617, $P = 2.39 \times 10^{-4}$), both associated with tumor progression[39,40] and metastatic potential[41] in PCa. In the Molecular Function category, enriched terms included *serine-type endopeptidase activity* (GO:0004252, $P = 2.70 \times 10^{-5}$), *serine-type peptidase activity* (GO:0008236, $P = 4.90 \times 10^{-5}$), *growth factor receptor binding*

(GO:0070851, $P = 7.63 \times 10^{-5}$), and *cytokine activity* (GO:0005125, $P = 1.17 \times 10^{-3}$), all relevant to tumor proliferation and microenvironmental signaling[42–44]. Cellular Component analysis indicated predominant localization to the *collagen-containing extracellular matrix* (GO:0062023, $P = 3.43 \times 10^{-5}$), highlighting roles in PCa development and metastasis[45] (Supplementary Data 4). Protein-protein interaction (PPI) analysis identified IL-10 (Interleukin-10) with the highest node degree of 18, followed by Angiostatin with a node degree of 14 (Supplementary Data 5), suggesting both topological centrality and functional significance in PCa.

Subsequent ingenuity pathway analysis (IPA) analysis was performed to explore potential regulatory mechanisms underlying the genes encoding the identified PCa-associated proteins (Supplementary Data 6). Kallikrein-related peptidase 3 (*KLK3*), (mitogen-activated protein kinase 3) (*MAPK3*), and ras-related protein (*RALB*) were significantly enriched in the PCa signaling pathway ($P = 7.76 \times 10^{-3}$). Additionally, several PCa-related biofunctions were identified, involving *DPEP1* (dipeptidase 1), *EGF*, *IGF2R* (insulin-like growth factor 2 receptor), *KLK3*, *LGALS3* (galectin−3), *LHCGR* (luteinizing hormone/chorionic gonadotropin receptor), *MAPK3*, *MSMB*, *PLG* (plasminogen), *SOCS3* (suppressor of cytokine signaling 3), *TLR3* (toll-like receptor 3). These biofunctions encompassed proliferation ($P = 8.42 \times 10^{-5}$), Cell cycle progression ($P = 5.91 \times 10^{-4}$), apoptosis ($P = 6.16 \times 10^{-4}$), binding ($P = 1.27 \times 10^{-3}$), PCa pathogenesis ($P = 2.94 \times 10^{-3}$), G0/G1 phase transition ($P = 3.48 \times 10^{-3}$), atrophy of prostate gland ($P = 3.48 \times 10^{-3}$), and hereditary PCa ($P = 3.48 \times 10^{-3}$) (Supplementary Data 6).

## Candidate drugs targeting identified associated proteins

To explore potential drug repurposing opportunities for the 79 uniquely identified proteins, we curated the DrugBank database and found 31 proteins as targets of FDA-approved drugs for various human diseases (Supplementary Data 7). Of these, 25 proteins were further supported as relevant to PCa based on a positive *overallAssociationScore* from the OpenTargets platform (Supplementary Data 7). Notably, the lutropin-choriogonadotropic hormone receptor (LSHR) is targeted by goserelin (DrugBank ID: DB00014), a synthetic analog of luteinizing hormone-releasing hormone used clinically to treat both breast cancer and PCa by reducing the secretion of pituitary gonadotropins[46,47]. Additionally, aldehyde dehydrogenase, mitochondrial (ALDH-E2) is targeted by disulfiram (DrugBank ID: DB00822), an FDA-approved drug for treating chronic alcoholism. In preclinical studies, disulfiram, along with delivering copper, targets and eradicates copper-laden PCa cells in animal models, while leaving non-cancer cells intact[48]. Our findings suggest promising opportunities for repurposing these drugs as potential treatments for PCa (Supplementary Data 7).

## Discussion

In this study, we developed comprehensive protein prediction models by integrating genetic and protein data collected from 1971 independent males representing diverse racial and ethnic backgrounds, including African, European, Asian, and Hispanic/Latino populations. Interestingly, the models for the Asian population exhibited the highest average cross-validated $R^2$ value of 0.18, despite this being the smallest subgroup within the MESA cohort. This suggests that, in this population, certain proteins are more strongly regulated by genetic variants, leading to improved predictive performance at specific loci. This hypothesis is further supported by the Kruskal−Wallis test, which revealed significant differences in heritability ($H^2$) values across the four groups ($\chi^2 = 156.89$, df = 3, $P < 2.20 \times 10^{-16}$). In post hoc Dunn's tests, $H^2$ values were significantly higher in Asians compared with Africans ($P = 0.02$), Europeans ($P = 4.38 \times 10^{-29}$), and Hispanic/Latino participants ($P = 2.69 \times 10^{-3}$). While these results are encouraging, they should be interpreted with caution given the limited sample size. Future studies leveraging larger and more diverse Asian

cohorts will be essential to validate and further investigate these observations.

These race- and ethnic-specific protein prediction models were then applied to GWAS summary statistics for PCa risk in each population of interest. Consequently, we identified three, 73, 15, and four proteins associated with PCa risk in African, European, Asian, and Hispanic/Latino populations, respectively. Furthermore, a trans-population meta-analysis detected 92 proteins that showed significant associations with PCa risk. To assess the robustness of causal inference, we applied the 2ScML method and found that over 40% of the identified population-specific associations remained statistically significant after accounting for potential pleiotropic effects of SNPs. Overall, the predicted abundances of a total of 104 proteins were associated with PCa risk in either population-specific or cross-population analyses. These proteins include targets such as cell adhesion molecules, protease inhibitors, receptors, enzymes, and cytokines, highlighting the multifaceted nature of the biological pathways involved in PCa development.

Among the proteins identified, several and/or their encoding genes have been previously reported to play crucial roles in PCa progression. For example, glucosamine-6 phosphate-N-acetyl Transferase (GNPNAT1) has been implicated in the pathogenesis of castration-resistant PCa[49] through the phosphatidylinositol 3-kinase/protein kinase B (PI3K/Akt) signaling pathway. Additionally, Shah et al. reported that enzalutamide-induced dysregulation of androgen receptor (AR) signaling increases exon-2 inclusion in PLA2G2A transcripts in PCa cells[50]. Moreover, PCa risk-associated SNPs rs9364554 and rs7629490 have been linked to the expression of nearby genes IGF2R and CHMP2B, respectively (±500 kb)[51]. We further compared our results to previously reported proteins associated with PCa risk in PWAS[18,19,52–54] and found 16 overlapping proteins (Supplementary Data 8). Of these, eight encoded by ARL3, B3GNT8, CTSS, IGF2R, MSMB, MICB, PLG, and SPINT2 have been demonstrated to be associated with PCa risk in our earlier studies, which included over 140,000 cases and controls of European descendant[18,19]. Among them, MSMB stands out as a well-established biomarker for PCa diagnosis and prognosis[32]. Two GWAS have reported the strongest association between PCa risk and rs10993994, located 2 base pairs upstream of the MSMB transcription start site[55,56]. This association has been validated in both European[57–59] and Asian populations[60,61]. Furthermore, two proteins identified in this study, encoded by ATF6B and PFKM, were previously implicated as potential causal genes at PCa susceptibility loci in our earlier transcriptome-wide association study[11].

In addition to proteins previously reported to be associated with PCa risk, our study also identified several additional proteins that may play important roles in PCa development. While some of these proteins have been linked to other cancer types, their potential involvement in PCa has not been previously characterized. For instance, LCTL encoding Lactase-like protein has been reported as a prognostic biomarker in glioma, where it modulates immune responses[62]. Moreover, SVEP1 expression is suppressed by miR-1269b, which activates the PI3K/Akt pathway and promotes recurrence and metastasis in hepatocellular carcinoma[63]. Our integrative analysis further identified PCa risk-associated proteins whose biological functions in prostate carcinogenesis remain largely unexplored. Functional annotation through IPA highlighted DPEP1 as a potential biomarker for PCa. DPEP1 has been shown to promote metastasis in colorectal cancer[64], but inhibit invasiveness in pancreatic ductal adenocarcinoma cells[65], suggesting a context-dependent role in tumor progression. These associations warrant further study to clarify their roles in PCa and may reveal new pathways or targets for biomarkers and therapy.

Through our drug repurposing analysis, we identified 31 proteins that are targets of approved drugs used to treat various human diseases, including PCa (treated by goserelin) (Supplementary Data 6). Among the list of implicated drugs, several drugs primarily used for non-PCa indications (alcohol dependence, inflammation, leukemia) demonstrate significant anti-PCa activity in preclinical models and in some cases early clinical settings, highlighting their potential for repurposing in PCa treatment. For example, in vitro and in vivo studies demonstrate that disulfiram (DrugBank ID DB00822), which targets ALDH-E2, has been shown to inhibit PCa cell growth, induces apoptosis, and demethylates DNA via DNMT1 inhibition[66,67]. Early-phase clinical trials in both recurrent[68] and metastatic PCa[69] have shown biological activity, although clinical benefit remains to be fully established. Additionally, mitogen-activated protein kinase 3 (ERK-1) is targeted by sulindac (DrugBank ID DB00605) and arsenic trioxide (DrugBank ID DB01169). Sulindac is a non-steroidal anti-inflammatory drug. Sulindac derivatives have demonstrated anti-proliferative and pro-apoptotic effects in PCa cell lines[70,71]. Arsenic trioxide, which is FDA-approved for the treatment of acute promyelocytic leukemia, has also shown promise in PCa. It induces apoptosis and cytotoxicity in PCa cell lines[72] and has demonstrated significant antitumor activity in an in vivo model of androgen-independent PCa[73]. Phase II clinical trial reported PSA reductions or disease stabilization in some patients with hormone-refractory PCa[74]. These findings underscore their promise for repurposing in PCa therapy and highlight the need for further clinical investigations to fully evaluate their efficacy and therapeutic value in this context.

One advantage of the current study is the use of genetic instruments rather than directly measured protein levels. Unlike traditional epidemiological approaches using measured protein levels for association estimation, our method minimized potential biases (including residual confounding, reverse causation, and selection bias) and also increased the statistical power. This improvement is achieved by applying comprehensive genetic prediction models to large-scale GWAS summary statistics of PCa risk, which involves a very large number of cases and controls. In our work, we also applied a design that leverages information from both cis- and trans- pQTLs to fully capture the genetically regulated components of protein levels. Although trans-regions exhibit more complex regulatory mechanisms than cis-regions, the majority of identified pQTLs are trans-acting instead of cis-acting[30]. Therefore, we integrated both cis- and trans-genetic instruments to develop our genetic prediction models for protein levels. The numbers of identified associations in African, Hispanic/Latino, and Asian populations were relatively smaller than that observed in European population. Future studies that increase the sample sizes of underrepresented groups including African, Hispanic/Latino, and Asian groups will be critical for enhancing our understanding of PCa etiology in these understudied populations and for reducing health disparity[24]. Another major strength of our study is the development of population-specific protein predicted models for males in African, Asian, and Hispanic/Latino groups. This method helps mitigate potential bias and loss of statistical power caused by varying linkage disequilibrium (LD) patterns across populations[29]. We leveraged a large blood proteome reference dataset to establish prediction models for nearly 7,000 proteins to examine their potential relationship with PCa.

Our study has several limitations that need to be acknowledged to appropriately interpret our findings. Firstly, we were unable to conduct external validation of the established prediction models for populations other than European population[30], due to a lack of data from additional well-designed studies. However, our external validation results focusing on the European population suggest that our modeling strategy is effective. As reference data of proteomic and genomic resources expand to include more globally underrepresented populations, this limitation can be well addressed in the near future. Secondly, the SomaScan platform has inherent limitations. Its aptamer-based detection method may be more susceptible to binding variability and off-target effects compared with antibody-based assays, such as Olink in the UK Biobank Pharma Proteomics Project (UKB-

PPP)[75]. Moreover, SomaScan interrogates only a subset of plasma proteins, and may not accurately reflect protein levels in other tissues or specific cell types. Future work using more comprehensive proteome platforms, as they become available, and extending investigations to other disease-relevant solid tissues and/or cell types will be crucial to better characterize additional disease-relevant proteins. Thirdly, due to the scope of the current study, we were unable to functionally characterize the potential roles of the identified proteins in PCa development. Future functional studies are warranted to elucidate the specific function and mechanisms of these proteins, particularly those not previously reported in prostate tumorigenesis.

In conclusion, our comprehensive PWAS, conducted across diverse populations and encompassing a large number of proteins (~7000), identified multiple associations between genetically predicted circulating levels of proteins and PCa risk. By developing population-specific genetic prediction models for protein abundances, we were able to elucidate distinct protein-PCa risk associations in each population of interest. These findings underscored the necessity of multi-population investigations to comprehensively elucidate the etiology of PCa and reduce health disparities.

## Methods

### Study population
The overall design of this study is illustrated in Fig. 3. The Multi-Ethnic Study of Atherosclerosis (MESA) included approximately 6500 men and women without known clinical cardiovascular disease at baseline, aged 45–84 years who were initially enrolled in 2000[76]. Baseline information and blood samples of the MESA participants were collected at their initial visit, which took place in six US states (New York, Maryland, Illinois, California, Minnesota, and North Carolina). Detailed

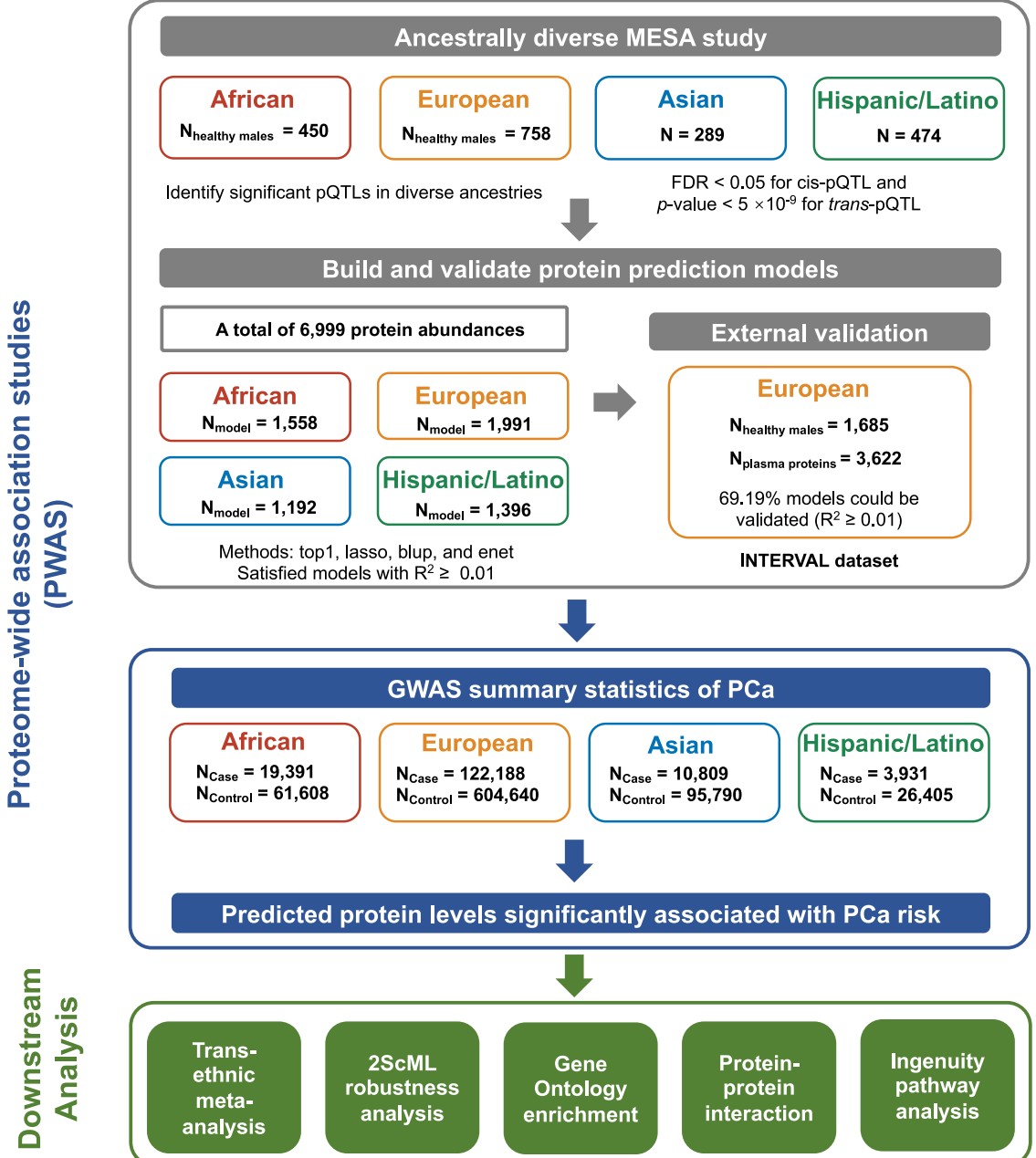

**Fig. 3 | The overall design of this study.** Key steps are illustrated with arrows connecting panels that indicate data processing, modeling, and analysis stages. Symbols and colors indicate different types of data and analytical procedures.

information on the MESA study design can be found elsewhere[76]. The MESA study protocol was reviewed and approved by the Institutional Review Boards (IRBs) of all participating institutions as well as by the National Heart, Lung, and Blood Institute (NHLBI). All participants provided written informed consent to participate in the parent study and received financial compensation.

In this study, we focused on 2013 male subjects who had no self-reported cancer diagnosis and no ICD-9 or ICD-10 cancer diagnoses in their hospitalization records or death certificates at baseline, including 453 African, 765 European, 296 Asian, and 499 Hispanic/Latino individuals residing in the USA. Population groups in our analysis were initially defined based on self-reported race/ethnicity as recorded in the original studies. All subjects had data available on blood protein levels (v1), genotype, and relevant covariates, including body mass index (BMI), sex, age, cigarette smoking status, and pack-years of cigarette smoking.

## Genotype data processing and quality control

The genotype data utilized in this study was generated using Affymetrix SNP 6.0, which was obtained from the MESA SNP Health Association Resource (SHARe) study (phs000420.v6.p3) and imputed on the Michigan imputation server (Minimac4.v1.0.0) using the 1000 Genomes reference panel[77]. For each population of interest, we excluded subjects who were identified as related via identity-by-descent (IBD) analysis, employing an in-house script that considered independent ($R^2 < 0.2$) and common (minor allele frequency (MAF) ≥ 5%) SNPs. This process resulted in a final sample of 1971 unrelated male subjects, including 450 of African background, 758 of European background, 289 of Asian background, and of 474 Hispanic/Latino background. Subsequently, SNPs were filtered based on pre-determined criteria including MAF > 0.05, genotyping missingness <5%, and adherence to Hardy-Weinberg equilibrium (HWE, $P > 5 \times 10^{-6}$) using PLINK v1.9 software. After filtering for variants present in the 1000 Genomes reference panel, a total of 8,011,863, 5,814,519, 5,308,357, and 6,058,753 high-quality SNPs were retained for African, European, Asian, and Hispanic/Latino populations, respectively. The LD-pruned ($R^2 < 0.2$), common (MAF ≥ 5%), and genotyped variants within 200 base pair windows were used to calculate genetic principal components (PCs) using the EIGENSOFT software[78].

## Proteomic data processing

Proteomic profiling was conducted using the aptamer-based SomaScan assay, which quantified 7289 human proteins. We excluded 10 SOMAmers whose target protein-encoding genes lacked positional information in the BioMart[79] database. An additional 280 SOMAmers targeting proteins encoded on sex chromosomes were removed to focus the analysis on plasma proteins or protein complexes encoded by autosomal genes. After these exclusions, 6999 proteins were retained for downstream analysis. The residuals of these 6999 protein abundances after adjusting for covariates were then transformed using a rank-based inverse normal transformation for model building.

## Building protein genetic prediction models

To identify informative SNPs for model building, we first conducted pQTL analyses adjusting for study site, BMI, sex, age, cigarette smoking status, pack-years of cigarette smoking, and top ten PCs. Significant cis-pQTLs were defined as SNPs within cis-regions associated with a protein at FDR < 0.05, while significant trans-pQTLs were defined as SNPs in trans-regions with $P < 5 \times 10^{-9}$. These significance thresholds were chosen to maximize the inclusion of potentially informative SNPs while minimizing excess noise[80]. We further extracted non-strand-ambiguous SNPs within 100 Kb of significant cis- and trans-pQTLs to serve as candidate predictors for each protein.

We used TWAS/FUSION framework[81] to construct subsequent genetic prediction models. Four methods were used for model construction: best linear unbiased predictor (BLUP), least absolute shrinkage and selection operator (LASSO), elastic net, and top SNPs (top1). BLUP estimates the joint effect sizes of all SNPs using a single variance component[82]. LASSO is a penalized regression method utilizing $L_1$ regularization techniques to produce sparse models[83]. As a generalization of the LASSO, elastic net linearly combines the $L_1$-penalty of LASSO and $L_2$-penalty of ridge regression, to select highly correlated variables together[84]. For each protein of interest, the prediction model with the most significant cross-validation P-value was retained. Only models with a cross-validation $R^2 > 0.01$ (indicating they explain more than 1% of the variance, corresponding to a minimum ~10% correlation between predicted and measured protein levels) were included in subsequent association analysis. This threshold is commonly applied in similar studies[8,80,85–88]. Cross-validation was performed using a five-fold scheme, where the dataset was randomly divided into five equal parts. In each fold, models were trained on 80% of the data and tested on the remaining 20%, rotating such that each subset served as a test set once. The final cross-validation performance was calculated by regressing observed protein levels against the predicted values aggregated from all test folds. This adjusted $R^2$ accounts for model complexity and sample size, providing a conservative measure of the variance explained by the genetic predictors.

We compared $R^2$ and $H^2$ across African, European, Asian, and Hispanic/Latino populations using the Kruskal–Wallis test, given the non-normal distribution of $R^2$ and $H^2$ values. When significant, post hoc pairwise comparisons were performed using Dunn's test with Bonferroni correction. Analyses were conducted in R (version 4.1.2) using the 'FSA' package.

## Validation of protein genetic prediction models of the European population

We conducted external validation of the European population models using data from 1685 healthy European males of the INTERVAL study, which measured plasma concentrations of 3622 proteins[30]. Genotype data were obtained using the Affymetrix Axiom UK Biobank genotyping array, and variants were phased with SHAPEIT3 before being imputed with a reference panel comprising the 1000 Genomes Phase 3-UK10K dataset. Log-transformed protein levels were adjusted for age, sex, duration between blood draw and processing, and the top three PCs[30]. The rank-inverse normalized residuals from the linear regression were used to compare with the predicted protein abundances, which were generated by applying the established genetic prediction models to the INTERVAL genetic data. Models with a performance $R^2$ value ≥ 0.01 (at least ~10% correlation between predicted and measured protein levels) were considered to have successfully passed external validation[18,80,86,89].

## Associations between genetically predicted circulating protein levels and PCa risk

We evaluated the associations between genetically predicted protein levels and PCa risk by leveraging the summary statistics of a large GWAS meta-analysis[4]. This meta-analysis included 156,319 PCa cases and 666,248 controls, comprising 19,391 PCa cases and 61,608 controls of African population, 122,188 cases and 604,640 controls of European population, 10,809 cases and 95,790 controls of Asian population, and 3931 cases and 26,405 controls of Hispanic/Latino population[4]. The TWAS/FUSION framework was used to evaluate the associations between predicted protein levels and PCa risk. The trans-population meta-analysis via an inverse variance fixed-effect approach was further implemented using METAL with fixed-effects model[90]. The protein-PCa risk associations were determined to be statistically significant with a threshold of FDR < 0.05. Evidence of heterogeneity of the associations across racial/ethnic groups was assessed using the $I^2$ statistic in METAL, with high heterogeneity defined as $I^2 > 75$ and a heterogeneity P-value (HetPVal) < 0.05[91].

## Robustness test using 2ScML method

The protein prediction models in this study were built utilizing the TWAS/FUSION software, which allows for the incorporation of multiple correlated SNPs as predictors via methods such as LASSO, BLUP, or elastic net. In actual analyses, the assumption of PWAS might be violated, as it assumes that all employed SNPs serve as valid instrumental variables. To assess the robustness of our main results, we applied the 2ScML method to infer the likely causal effects of the associated proteins on PCa risk. This method relies on a plurality condition, which posits that the largest cluster of SNPs sharing the same causal effect comprises valid instrumental variables, thereby permitting more than 50% of the SNPs to be invalid[92]. By accounting for the horizontal pleiotropic effects of the SNPs in the protein prediction models, 2ScML identifies valid instruments through constrained maximum likelihood estimation. This method is applicable when at least three predictors are present, enabling the selection of potentially pleiotropic SNPs. A threshold of FDR < 0.05 was applied to determine significant associations using the 2ScML method.

## Tissue expression, functional enrichment, and network analysis

To evaluate tissue-specific expression, we cross-referenced 94 PCa–associated genes with RNA expression data from the HPA[93]. Genes were classified based on HPA criteria as: (1) Elevated in prostate, (2) Elevated in other but expressed in prostate, (3) Low tissue specificity but expressed in prostate, (4) Not detected in prostate, (5) Not detected in any tissue. Classifications were confirmed using prostate-specific RNA expression levels, enabling assessment of prostate relevance and transcriptional activity of each gene.

To investigate the biological functions of 96 unique proteins encoded by 94 genes identified in the PWAS, we performed GO enrichment analysis across three domains: Biological Process, Molecular Function, and Cellular Component. GO enrichment was conducted using 'ClusterProfiler' package[94], with the human genome (org.Hs.eg.db) as background. Only terms with adjusted P-value < 0.05 were considered significant.

Proteins operate within a complex network of intermolecular interactions rather than acting independently[95]. To understand the networks of PCa risk-associated proteins, we constructed a PPI network using the STRING database (https://string-db.org/, accessed on June 18, 2025). Additionally, we employed IPA (version 03-29-25) to perform an enrichment analysis of the genes encoding identified proteins, assessing their enrichment in canonical pathways, molecular and cellular functions, and networks. Detailed methodology for this tool has been described previously[96].

## Drug repurposing analysis

To explore potential drug repurposing opportunities, we queried the DrugBank database[97] to investigate evidence of existing drugs targeting the identified associated proteins of interest. This analysis may reveal promising candidate drugs for further investigation of their potential efficacy in PCa treatment. We further assessed the potential relevance of these proteins to PCa using data from OpenTargets[98]. Specifically, proteins with a positive *overallAssociationScore* for PCa-related outcomes, including prostate-specific antigen levels, prostate carcinoma, prostate adenocarcinoma, and PCa, were retained for further consideration.

## Reporting summary

Further information on research design is available in the Nature Portfolio Reporting Summary linked to this article.

## Data availability

Specific genome, proteome, and covariate data of MESA[76] have been deposited to the database of Genotypes and Phenotypes (dbGaP) under accession code phs000209.v13.p3 (https://www.ncbi.nlm.nih. gov/projects/gap/cgi-bin/study.cgi?study_id=phs000209.v13.p3). Additional data of MESA are available through a concept proposal application via MESA Genetics P and P Committee. These data are available under restricted access to protect participant privacy and comply with informed consent agreements. For data available through dbGaP, access is limited to qualified researchers who submit a Data Access Request (DAR) through dbGaP. Such requests must include a research use statement and a data use certification signed by the principal investigator and the institutional signing official. Requests will be reviewed by the relevant Data Access Committee (DAC) with a timeline developed by dbGaP. Once approved by dbGaP, access will be granted for one year. It requires an annual renewal via dbGaP to continue access beyond the initial 12-month period. Individual level data of genotype and proteomic data of INTERVAL[30] study are available under controlled access in European Genome-phenome Archive (EGA) under accession number EGAS00001002555. Access is restricted to protect participant privacy and is granted to qualified researchers following approval by the EGA Data Access Committee. Researchers interested in accessing these data can submit a request via the EGA data access portal, and access can be provided after the EGA Data Access Committee reviews and approves it. DAC Aim to respond to all initial requests in less than 2 weeks. The length of time you can access the data depends on the terms set by the DAC. The publicly available summary statistics of multi-population PCa GWAS[4] are available on the GWAS Catalog (https://www.ebi.ac.uk/gwas/). These statistics are categorized by different racial/ethnicity groups with the following accession codes: European (GCST90274714, https://www.ebi.ac.uk/gwas/studies/GCST90274714)[4], African (GCST90274715, https://www.ebi.ac.uk/gwas/studies/GCST90274715)[4], Asian (GCST90274716, https://www.ebi.ac.uk/gwas/studies/GCST90274716)[4], and Hispanic/Latino (GCST90274717, https://www.ebi.ac.uk/gwas/studies/GCST90274717)[4]. The remaining data are available within the Article, Supplementary Information or Source Data file. Source data are provided with this paper.

## Code availability

The code used to perform the analyses in this study is available at https://github.com/HuaZ-bioinfomatics/MESA-7K-PWAS-PCa and Zenodo https://zenodo.org/records/17280225[99].

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

## Acknowledgements

The authors also would like to thank all of the individuals for their participation in the parent studies and all the researchers, clinicians, technicians and administrative staff for their contribution to the studies. The contents of this paper are solely the responsibility of the authors and do not necessarily represent the official views of the NIH. Lang Wu is supported by National Human Genome Research Institute/National Institute on Minority Health and Health Disparities (NHGRI/NIMHD) U54 HG013243 and National Cancer Institute R01CA263494 and U01CA293883. Hua Zhong and Shuai Liu are partially supported under award number U24DK132746-01, UCLA LIFT-UP (Leveraging Institutional support for Talented, Upcoming Physicians and/or Scientists). Peter Ganz, Rajat Deo, and Ruth F Dubin are supported by R01HL159081. The MESA projects are conducted and supported by the National Heart, Lung, and Blood Institute (NHLBI) in collaboration with MESA investigators. Support for MESA is provided by contracts 75N92020D00001, HHSN268201500003I, N01-HC-95159, 75N92020D00005, N01-HC-95160, 75N92020D00002, N01-HC-95161, 75N92020D00003, N01-HC-95162, 75N92020D00006, N01-HC-95163,75N92020D00004, N01-HC-95164, 75N92020D00007, N01-HC-95165, N01-HC-95166, N01-HC-95167, N01-HC-95168, N01-HC-95169, UL1-TR-000040, UL1-TR-001079, UL1-TR-001420, UL1TR001881, DK063491, and R01HL105756. The authors thank the other investigators, the staff, and the participants of the MESA study for their valuable contributions. A full list of participating MESA investigators and institutes can be found at http://www.mesa-nhlbi.org.

## Author contributions

L.W. conceived the study and supervised the work. J.Z. contributed to the study design and conducted drug repurposing analysis. S.L., H.Z., and J.Z. performed statistical analyses. H.Z. performed PPI and IPA analyses. H.Z., J.Z, S.L., and L.W. wrote the initial draft manuscript. C.W., L.W., S.P.W., C.H.M., M.J.B., P.D., X.G., C.W.J., H.J.L., K.D.T., R.P.T., R.Y., A.W.M., S.S.R., J.I.R., R.D., R.F.D., and P.G. conducted the experiment, contributed materials, and provided valuable feedback for revising the manuscript. All authors have reviewed and approved the final manuscript.

## Competing interests

Lang Wu provided consulting services to Pupil Bio Inc., Techspert, and Galiher DeRobertis & Waxman LLP, and reviewed manuscripts for *Gastroenterology Report*, not related to this study, and received honorarium. No potential conflicts of interest were disclosed by the other authors.

## Additional information

[1]Cancer Epidemiology Division, Population Sciences in the Pacific Program, University of Hawai'i Cancer Center, University of Hawai'i at Mānoa, Honolulu, HI, USA. [2]Department of Interdisciplinary Oncology and Department of Genetics, LSU-LCMC Health Cancer Center, School of Medicine, Louisiana State University Health Sciences Center, New Orleans, LA, USA. [3]Department of Quantitative Health Sciences, John A. Burns School of Medicine, University of Hawai'i at Mānoa, Honolulu, HI, USA. [4]Department of Biostatistics, The University of Texas MD Anderson Cancer Center, Houston, TX, USA. [5]Department of Tumor Microenvironment and Metastasis, Moffitt Cancer Center, Tampa, FL, USA. [6]Johns Hopkins Ciccarone Center for the Prevention of Cardiovascular Disease, Johns Hopkins School of Medicine, Baltimore, MD, USA. [7]Department of Oncology, Sidney Kimmel Comprehensive Cancer Center, Johns Hopkins University, Baltimore, MD, USA. [8]Department of Medicine, Johns Hopkins Hospital, Baltimore, MD, USA. [9]Laboratory for Clinical Biochemistry Research, University of Vermont, Burlington, VT, USA. [10]The Institute for Translational Genomics and Population Sciences, Department of Pediatrics, The Lundquist Institute for Biomedical Innovation at Harbor-UCLA Medical Center, Torrance, CA, USA. [11]Collaborative Health Studies Coordinating Center, University of Washington, Seattle, WA, USA. [12]Division of Cardiovascular Sciences, Epidemiology Branch, National Heart, Lung and Blood Institute, Bethesda, MD, USA. [13]Department of Genome Sciences, University of Virginia, Charlottesville, VA, USA. [14]Division of Cardiovascular Medicine, Perelman School of Medicine at the University of Pennsylvania, Philadelphia, PA, USA. [15]Department of Medicine, University of Texas Southwestern Medical Center, Dallas, TX, USA. [16]Division of Cardiology, Zuckerberg San Francisco General Hospital and Department of Medicine, University of California San Francisco, San Francisco, CA, USA. [17]These authors contributed equally: Hua Zhong, Jingjing Zhu. ✉e-mail: lwu3@lsuhsc.edu

