## [Transparent Peer Review file · Nature Communications]

Proteome-wide association study of prostate cancer risk across populations

Corresponding Author: Dr Lang Wu

Version 0:

Reviewer comments:

Reviewer #1

(Remarks to the Author)

The authors report an important proteome-wide association study (PWAS) of prostate cancer (PCa) risk utilizing proteomics reference dataset of MESA, the currently most comprehensive dataset that includes the highest number of proteins in the field. The study focuses on PCa, a common malignancy that brings huge public health burden. The findings of this study provide substantial new insights into etiology of this common cancer, which can potentially facilitate future development of novel therapeutic strategies and risk assessment tools for this cancer. The study itself generates comprehensive protein genetic prediction models that can significantly help uncover novel and causally relevant proteins for multiple other human diseases, beyond the scope of PCa. Authors have agreed that they will deposit the models to a public domain, which can significantly enhance the utility for other researchers in the field. The manuscript is generally well written. The analytical strategy is rigorous and appropriate. PWAS is very promising to dissect the genetic architecture of complex disease and risk, including cancer. This study is quite innovative in the field and the findings will substantially impact on the genetic study of prostate cancer in future. I have only several minor methodological and interpretative suggestions for the authors to improve the clarity and further enhance the overall impact of the study.

1. The authors report using models trained on MESA proteomics data to impute protein levels into PCa GWAS summary statistics. However, the manuscript lacks a detailed presentation of the protein prediction model performance metrics (e.g., cross-validated R^2 , number of SNPs used per model etc). These are information useful. I recommend including a supplementary table summarizing these metrics for all tested proteins.
2. While the manuscript reports specific significant PWAS association signals/protein loci, it is not much clear how many of these proteins map to the loci previously identified in PCa GWAS (genome-wide association studies). An explicit analysis comparing the PWAS signals to known GWAS hits would help differentiate novel findings and more clearly articulate the added value of the current study/findings.
3. The manuscript would benefit from a more thorough comparison to existing transcriptome-wide association studies (TWAS) or previously published PWAS results in PCa. To have a more in-depth discussion of such overlaps—and highlighting genes/proteins uniquely discovered in the current study—would better contextualize the findings and help clarify the novel contribution of this work.
4. While the computational analyses are well executed and sufficient to support their findings/conclusions, the manuscript somewhat lacks a deeper biological interpretation of the identified protein associations. The authors are encouraged to incorporate:
 - o Functional enrichment or pathway analyses to identify whether associated proteins participate in coherent biological mechanisms;
 - o Expression data to assess whether these proteins are relevantly expressed in prostate tissue;
 - o Integration with public experimental datasets (e.g., CRISPR screens, Open Targets, and DepMap) to assess potential functional impact.

(Remarks on code availability)

There are five R codes. They appear appropriate.

Reviewer #2

(Remarks to the Author)

The manuscript by Zhong et al. describes a large proteome-wide association study of prostate cancer risk in 4 populations: African, Asian, European, and Latin/Hispanic using the MESA cohort. The model generated for Europeans was validated using INTERVAL. Overall, the analysis identified 2 (African), 3 (Hispanic), 7 (Asian) and 65 (European) proteins significantly associated with PCa risk; a trans-population meta-analysis reported 83 PCa-risk associated proteins. To address potential clinical significance, the investigators examined drug repurposing opportunities of the identified proteins. This is the largest study of its kind in PCa, generating interesting and informative results to the greater field.

Overall, the manuscript is carefully analyzed and the data are clearly presented and organized in a manner that is easy to follow. The figures are well-labeled and described. My only concerns speak to the clinical significance of the study, as well as how the 4 populations were determined. Specifically:

1. Would it be possible to examine aggressive PCa-risk? This is more clinically impactful, particularly when looking at drug repurposing.
2. How were the population groups determined? Self-reported race or via GWAS-estimated markers of ancestry? Given the focus on different population groups, I looked for (and could not find) how these categories were created.
3. There were points in the results section of the manuscript that were noted as "notable" (see line 153) but no further discussion was included. Why did the prediction models for Asian populations have the highest predictive value (especially since this was the smallest MESA population).

(Remarks on code availability)

Reviewer #3

(Remarks to the Author)

This study conducted a large-scale proteome-wide association study (PWAS) to explore the link between genetically predicted circulating protein levels and prostate cancer (PCa) risk across diverse populations. Using data from the MESA study, researchers developed genetic models for thousands of plasma proteins in African, European, Asian, and Hispanic/Latino men. Summary statistics from over 900,000 individuals were utilized to assess associations of genetically predicted proteins with PCa risk. This is an important study given that investigation of plasma proteins with PCa risk has been limited, especially in non-European populations.

Below are my comments:

1. Incomplete sentence on page 4 line 105.
2. Why only 6,999 out of 7,367 plasma proteins were tested (page 6 line 141)?
3. What was the rationale of using a FDR of 0.05 for cis-pQTLs while trans-pQTLs were identified at $P < 5 \times 10^{-9}$?
4. How do you determine the cutoff of performance $R^2 \geq 0.01$ for external validation?
5. Is it a typo on page 19 line 442?
6. In Figure 3, it would be helpful to indicate whether the N was for number of subjects or for number of proteins.
7. The authors mentioned that the models with a cross-validation $R^2 > 0.01$ were used for further association analysis. It was not clear how the cross-validation was conducted within each population. Also, the cross-validation $R^2 > 0.01$ appears to be low. The same cutoff was used for external validation for European population. Why not focus on models with a higher R^2 to focus on proteins with better genetic predictions which could also improve the power of this study?

(Remarks on code availability)

I do not see any codes provided.

Version 1:

Reviewer comments:

Reviewer #1

(Remarks to the Author)

The authors have fully addressed all my comments. This reviewer appreciated their major efforts to compare with other types of evidence (PCa GWAS, TWAS, etc.) and further interpretation. The new results solidified their results, making their conclusion more reliable. Therefore, the revised manuscript has had many improvements for high quality. I do not have any further comments except for one minor suggestion below, which I believe the authors can address well in the final version (without my further review).

1. I appreciated the authors provided a detailed Suppl Table 1. How are these rows ordered? In addition, Column "Target" is for protein and Column "EntrezGeneSymbol" is for gene symbols. Many protein abbreviations in Target column are based on protein symbols. The authors may double check to see some of abbreviations can be better by using protein names, or provide the source of such protein abbreviations in the table footnote.

Reviewer #2

(Remarks to the Author)

The authors responded appropriately to the reviewer concerns. Nicely done.

Reviewer #3

(Remarks to the Author)

The authors have adequately addressed the comments from previous review. No further comments.

RE: NCOMMS-25-32305-T

Title: Proteome-wide association study of prostate cancer risk in racial and ethnic diverse populations

Reviewer #1

Comment 1: *The authors report using models trained on MESA proteomics data to impute protein levels into PCa GWAS summary statistics. However, the manuscript lacks a detailed presentation of the protein prediction model performance metrics (e.g., cross-validated R², number of SNPs used per model etc). These are information useful. I recommend including a supplementary table summarizing these metrics for all tested proteins.*

Response-1:

We thank the reviewer for this helpful suggestion. We agree that providing the performance metrics of the protein prediction models is important. In the revised manuscript, we have now added a new supplementary table (**Supplementary Table S1**) that summarizes key metrics for all tested proteins, including the cross-validated R² and the number of SNPs (*cis*- and *trans*-) used in each model.

Page 5, Lines 165-168:

We further established prediction models for 1,578 proteins in the African population, 1,993 in the European population, 1,218 in the Asian population, and 1,390 in the Hispanic/Latino population, each with an R² ≥ 0.01, with 697 proteins overlapping across all genetic ancestries (Figure 1a and Table S1)

Table S1. Characteristics of the Established Prediction Models with R²>0.01 (part)

Table S1. Characteristics of the Established Prediction Models with R ² >0.01.											
model	ID	Somaid	Target Full Name	Target	EntrezGeneSymbol	best_model	Req	N_SNP	N_cis	N_trans	race
seq_10012.5.wgt.RDat	seq_10012.5	SL014669	SAM pointed domain-containing Ets transcription factor	SPDEF	SPDEF	blup	0.07	469	0	469	AFA
seq_10037.99.wgt.RDat	seq_10037.99	SL025986	Sialic acid-binding Ig-like lectin 12: Ig-like V-type 2 domain, isoform long	SIG12: Ig-like V-type 2	SIGLEC12	lasso	0.29	9	9	0	AFA
seq_10046.55.wgt.RDat	seq_10046.55	SL003727	Baculoviral IAP repeat-containing protein 2	clAP-1	BIRC2	lasso	0.42	5	0	5	AFA
seq_10047.12.wgt.RDat	seq_10047.12	SL004553	Neutrophil cytosol factor 2	NCF-2	NCF2	top1	0.11	1	0	1	AFA
seq_10048.7.wgt.RDat	seq_10048.7	SL011998	Core-binding factor subunit beta	PEBB	CBFB	top1	0.32	1	0	1	AFA
seq_10054.3.wgt.RDat	seq_10054.3	SL019232	Gigaxonin	GAN	GAN	top1	0.17	1	0	1	AFA
seq_10082.251.wgt.RDat	seq_10082.251	SL019251	Neurofilament light polypeptide	NFL	NEFL	lasso	0.43	16	0	16	AFA
seq_10339.48.wgt.RDat	seq_10339.48	SL001731	Gamma-enolase	NSE	ENO2	top1	0.05	1	1	0	AFA
seq_10344.334.wgt.RDat	seq_10344.334	SL005173	Interleukin-10 receptor subunit alpha	IL-10 Ra	IL10RA	blup	0.06	416	0	416	AFA
seq_10346.5.wgt.RDat	seq_10346.5	SL007221	Signal transducer and activator of transcription 3	STAT3	STAT3	blup	0.01	802	802	0	AFA
seq_10365.132.wgt.RDat	seq_10365.132	SL005184	Interleukin-23	IL-23	IL12B/IL23A	enet	0.03	34	34	0	AFA
seq_10366.11.wgt.RDat	seq_10366.11	SL001728	Platelet-derived growth factor receptor alpha	PDGFRA	PDGFRA	blup	0.02	1220	1159	61	AFA
seq_10382.1.wgt.RDat	seq_10382.1	SL006803	Angiotensin-related protein 3	ANGL3	ANGPTL3	blup	0.08	1563	1514	49	AFA
seq_10391.1.wgt.RDat	seq_10391.1	SL006803	Angiotensin-related protein 3	ANGL3	ANGPTL3	top1	0.03	1	1	0	AFA
seq_10419.1.wgt.RDat	seq_10419.1	SL014110	Scavenger receptor class A member 5	SCAR5	SCAR5	top1	0.16	1	1	0	AFA
seq_10439.57.wgt.RDat	seq_10439.57	SL005393	Alpha-amylase 2B	Alpha-amylase 2B	AMY2B	blup	0.04	105	105	0	AFA
seq_10440.26.wgt.RDat	seq_10440.26	SL025782	CXADR-like membrane protein: Extracellular domain	CLMP	ACAM5CD	top1	0.04	1	1	0	AFA
seq_10442.1.wgt.RDat	seq_10442.1	SL018089	Transmembrane protein 190	TM190	TMEM190	top1	0.36	1	1	0	AFA

Comment 2: *While the manuscript reports specific significant PWAS association signals/protein loci, it is not much clear how many of these proteins map to the loci previously identified in PCa GWAS (genome-wide association studies). An explicit analysis comparing the PWAS signals to known GWAS hits would help differentiate novel findings and more clearly articulate the added value of the current study/findings.*

Response-2:

We thank the reviewer for this thoughtful comment. To clarify the relationship between our PWAS findings and previously reported PCa GWAS loci (e.g., Conti et al., 2021; Wang et al., 2023), we have now conducted an explicit comparison between the significant PWAS signals and known GWAS risk loci for PCa in the Results section (Page 6, Lines 397-403). We summarized the

Comment 4: While the computational analyses are well executed and sufficient to support their findings/conclusions, the manuscript somewhat lacks a deeper biological interpretation of the identified protein associations. The authors are encouraged to incorporate:

o Functional enrichment or pathway analyses to identify whether associated proteins participate in coherent biological mechanisms;

o Expression data to assess whether these proteins are relevantly expressed in prostate tissue;

o Integration with public experimental datasets (e.g., CRISPR screens, Open Targets, and DepMap) to assess potential functional impact.

Response-4:

We thank the reviewer for these insightful suggestions. We agree that adding biological context would further strengthen the manuscript. We have now undertaken several additional analyses to provide deeper interpretation of the identified protein associations:

• **Functional enrichment and pathway analyses:** We performed Gene Ontology (GO) and pathway enrichment analyses using the significant PWAS-identified proteins. The results highlight several biologically coherent pathways relevant to PCa, including the *insulin-like growth factor receptor signaling pathway* (GO:0048009, $P = 2.92 \times 10^{-5}$) and *extracellular matrix disassembly* (GO:0022617, $P = 2.39 \times 10^{-4}$). These findings are now included in the Results section of the revised manuscript on Page 7, Lines 602-613 and detailed in **Supplementary Table S4**.

• **Expression in prostate tissue:** We systematically examined expression profiles of the identified proteins using Human Protein Atlas data to assess their expression levels in prostate tissue. We found that three of the genes encoding associated proteins (*KLK3*, *MSMB*, and *ALOX15B*) exhibited prostate-enriched expression. These well-known PCa markers validate our gene selection strategy. An additional 56 genes encoding the identified proteins also remained measurably expressed in the prostate tissue, many of which have been previously implicated in PCa-related pathways, including growth factor signaling, extracellular matrix remodeling and metastasis, and inflammatory regulation. These findings are now included in the Results section of the revised manuscript on Page 7, Lines 594-601 and detailed in the newly added column, **“Expression specificity and distribution,”** in **Supplementary Table S2**.

• **Integration with public experimental datasets:** To further assess functional relevance, we integrated our findings with data from large-scale experimental resources via the OpenTargets platform. We identified 25 of 31 proteins with a positive *overallAssociationScore* for PCa-related outcomes, supporting their relevance to PCa. These results are now described in the Results section of the revised manuscript (Page 8, Lines 685–687) and detailed in the newly added column, **“Overall association score,”** in **Supplementary Table S7**.

Page 7, Lines 593-613:

Tissue expression, functional enrichment, and network analysis of identified proteins

We evaluated the tissue-specific expression of 94 genes coding the 96 proteins identified in the PWAS by cross-referencing RNA expression data from the Human Protein Atlas. Among these,

three genes (*KLK3*, *MSMB*, and *ALOX15B*) exhibited prostate-enriched expression. These well-known *PCa* markers ³²⁻³⁴, validates our gene selection strategy. An additional 56 genes showed higher expression in other tissues but remained measurably expressed in the prostate, many of which have been previously implicated in *PCa*-related pathways, including growth factor signaling (e.g. *EGF*³⁵, *IGFBP3*³⁶), extracellular matrix remodeling and metastasis (e.g. *PRSS3*³⁷, *MMP7*³⁸), and inflammatory regulation (e.g. *SOCS3*³⁹).

Our Gene Ontology (GO) enrichment analysis of 94 genes coding 96 *PWAS*-identified proteins revealed significant overrepresentation in several biologically relevant categories (Table S4). In the Biological Process domain, key enriched terms included insulin-like growth factor receptor signaling pathway (GO:0048009, $P = 2.92 \times 10^{-5}$) and extracellular matrix disassembly (GO:0022617, $P = 2.39 \times 10^{-4}$), both previously associated with tumor progression^{32,33} and metastatic potential³⁴ in *PCa*. In the Molecular Function category, enriched terms included serine-type endopeptidase activity (GO:0004252, $P = 2.70 \times 10^{-5}$), serine-type peptidase activity (GO:0008236, $P = 4.90 \times 10^{-5}$), growth factor receptor binding (GO:0070851, $P = 7.63 \times 10^{-5}$), and cytokine activity (GO:0005125, $P = 1.17 \times 10^{-3}$), all of which are implicated in tumor proliferation and microenvironmental signaling ³⁵⁻³⁷. For Cellular Component, proteins were predominantly localized to the collagen-containing extracellular matrix (GO:0062023, $P = 3.43 \times 10^{-5}$), indicating roles in *PCa* development and metastasis ³⁸ (Table S4).

Page 8, Lines 685-687:

Of these, 25 proteins were further supported to be relevant to *PCa*, as indicated by an annotated overallAssociationScore greater than zero for *PCa*-related outcomes based on interrogation through the OpenTargets platform (Table S7).

Table S2. Significant protein-PCa associations identified in our study (part).

ID	Target Full Name	Target	Enzyme Class Symbol	Protein properties			Molecular Weight			pI			Isoelectric Point			Association in Population	where 0=MS of	Homoel_GWAA_SNP	Distance_to_SNP	Expression specificity and distribtu	
				UniProt	RefSeq	Ensembl	FW	MOLEC	FW	MOLEC	pI	FW	MOLEC	pI	FW						MOLEC
ms15012	Brain neurotrophin-3	BDNF	MBLH	162	17.04	17.01	17.01	17.01	17.01	17.01	17.01	17.01	17.01	17.01	17.01	17.01	17.01	17.01	17.01	17.01	17.01
ms15013	Brain neurotrophin-4	BDNF	MBLH	162	17.04	17.01	17.01	17.01	17.01	17.01	17.01	17.01	17.01	17.01	17.01	17.01	17.01	17.01	17.01	17.01	17.01
ms15014	Brain neurotrophin-5	BDNF	MBLH	162	17.04	17.01	17.01	17.01	17.01	17.01	17.01	17.01	17.01	17.01	17.01	17.01	17.01	17.01	17.01	17.01	17.01

Table S4. GO enrichment analysis of genes encoding candidate proteins (part).

Biological Process	ID	Description	GeneRatio	BgRatio	pvalue	p.adjust	qvalue	geneID	Count
GO:0048009	GO:0048009	insulin-like grc 4/90	37/18723		2.92E-05	0.018511982	0.016576532	57462/3486/1	4
GO:0060670	GO:0060670	branching invr 3/90	13/18723		2.97E-05	0.018511982	0.016576532	10653/9021/3	3
GO:0048608	GO:0048608	reproductive ; 10/90	424/18723		3.63E-05	0.018511982	0.016576532	10653/15144	10
GO:0061458	GO:0061458	reproductive ; 10/90	427/18723		3.85E-05	0.018511982	0.016576532	10653/15144	10
GO:0032091	GO:0032091	negative regul 5/90	94/18723		8.99E-05	0.034566167	0.030952234	4043/5595/35	5
GO:0060713	GO:0060713	labyrinthine la 3/90	22/18723		0.000154816	0.038109935	0.034125497	10653/9021/3	3
GO:0005976	GO:0005976	polysaccharid 5/90	107/18723		0.000165832	0.038109935	0.034125497	1950/5834/37	5
GO:0043567	GO:0043567	regulation of ir 3/90	24/18723		0.000202062	0.038109935	0.034125497	57462/3486/1	3
GO:0032609	GO:0032609	interferon-gan 5/90	112/18723		0.000205365	0.038109935	0.034125497	149233/7098/	5
GO:0032649	GO:0032649	regulation of ir 5/90	112/18723		0.000205365	0.038109935	0.034125497	149233/7098/	5
GO:0002697	GO:0002697	regulation of ir 8/90	339/18723		0.000227005	0.038109935	0.034125497	354/3958/259	8
GO:0022617	GO:0022617	extracellular n 4/90	63/18723		0.000239459	0.038109935	0.034125497	5340/1520/25	4
GO:0060669	GO:0060669	embryonic ple 3/90	26/18723		0.000257768	0.038109935	0.034125497	10653/9021/3	3
Molecular Function	ID	Description	GeneRatio	BgRatio	pvalue	p.adjust	qvalue	geneID	Count
GO:0004252	GO:0004252	serine-type er 7/92	174/18368		2.70E-05	0.004948978	0.004151893	354/5340/564	7
GO:0008236	GO:0008236	serine-type pi 7/92	191/18368		4.90E-05	0.004948978	0.004151893	354/5340/564	7
GO:0017171	GO:0017171	serine hydrol 7/92	195/18368		5.58E-05	0.004948978	0.004151893	354/5340/564	7
GO:0070851	GO:0070851	rowth factor 6/92	141/18368		7.63E-05	0.005072557	0.004255568	1950/27179/1	6

Table S7. Drug repurposing opportunities of identified proteins (part).

ID	Target Full Name	Target	Gene Symbol	Overall association score	Drugbank ID	Drug name	Cas number	Drug group	Drug type	Molecular action
seq.12571.14	ADP-ribosylation factor-like protein 3	ARL3	ARL3	0.25*	DB03814	2-(N-morpholino)ethanesulfonic acid	4432-31-9	experimental	small molecule	
seq.17341.89	Acetyl-CoA acetyltransferase,	THIC	ACAT2	0.24**	DB04315	Guanosine-5'-Diphosphate	146-91-8	experimental	small molecule	
					DB01915	S-Hydroxycysteine		experimental	small molecule	
					DB01992	Coenzyme A	85-61-0	investigational, nutraceutical	small molecule	
					DB00157	NADH	58-68-4	approved, nutraceutical	small molecule	
					DB00536	Guanidine	113-00-8	approved	small molecule	inhibitor
					DB00822	Disulfiram	97-77-8	approved	small molecule	inhibitor
seq.18381.16	Aldehyde dehydrogenase, mitochondrial	ALDH-E2	ALDH2	0.19***	DB02115	Daidzin	552-66-9	experimental	small molecule	inhibitor
					DB04202	Isoformononein		experimental	small molecule	inhibitor
					DB04381	Crotonaldehyde	123-73-9	experimental	small molecule	inhibitor
					DB09116	Calcium carbimide	156-62-7	approved, withdrawn	small molecule	inhibitor
					DB14128	Nadide	53-84-9	experimental	small molecule	inhibitor

Reviewer #2

Comment 1: Would it be possible to examine aggressive PCa-risk? This is more clinically impactful, particularly when looking at drug repurposing.

Response-1:

We thank the reviewer for this thoughtful suggestion. We agree that focusing on aggressive PCa risk would enhance the clinical relevance of our findings, particularly in the context of drug repurposing. The goal of the current study is to identify proteins associated with PCa susceptibility. We plan to pursue the analyses for aggressiveness in a separate study, by closely working with colleagues in the PCa consortia.

Comment 2: How were the population groups determined? Self-reported race or via GWAS-estimated markers of ancestry? Given the focus on different population groups, I looked for (and could not find) how these categories were created.

Response-2:

We thank the reviewer for pointing this out. The population groups in our analysis were determined based on self-reported race/ethnicity as collected in the original studies.

We have now clarified this in the Methods section of the manuscript on Pages 12-13, Lines 976-978. We believe this additional detail will help readers better understand how population groups were defined and validated in our analysis.

Pages 12-13, Lines 976-978:

Population groups in our analysis were initially defined based on self-reported race/ethnicity as recorded in the original studies.

Page 13, Lines 995-997:

The LD-pruned ($R^2 < 0.2$), common ($MAF \geq 5\%$), and genotyped variants within 200 base pair windows were used to calculate genetic principal components (PCs) using the EIGENSOFT software⁸¹.

Comment 3: There were points in the results section of the manuscript that were noted as “notable” (see line 153) but no further discussion was included. Why did the prediction models for Asian populations have the highest predictive value (especially since this was the smallest MESA population).

Response-3:

We thank the reviewer for this great suggestion. Using the Kruskal–Wallis test, we found a significant difference in cross-validated R^2 values of the established protein genetic prediction models across the four groups ($\chi^2 = 98.4$, $df = 3$, $p < 2.2 \times 10^{-16}$). Post hoc Dunn’s tests with Bonferroni correction revealed that R^2 values were significantly higher in Asian population compared with African ($p < 2.6 \times 10^{-5}$), European ($p < 7.8 \times 10^{-22}$), and Hispanic/Latino populations ($p < 1.3 \times 10^{-5}$). These findings are now included in the Results section of the revised manuscript on Page 5, Lines 174-177.

The higher average predictive performance observed in the Asian population, despite its smaller sample size in MESA, is interesting. We assessed whether certain proteins may be under stronger genetic regulation (e.g., higher heritability) in this group, which could enhance performance of corresponding genetic prediction models at specific loci. The Kruskal–Wallis test also revealed significant differences in heritability (H^2) across the four groups ($\chi^2 = 156.89$, $df = 3$, $p < 2.20 \times 10^{-16}$). Post hoc Dunn’s tests showed that H^2 values were significantly higher in Asians compared with Africans ($p = 0.02$), Europeans ($p = 4.38 \times 10^{-29}$), and Hispanic/Latino participants ($p = 2.69 \times 10^{-3}$). If validated in additional studies, this interesting finding may reveal unique genetic regulation pattern in Asian population that can be further investigated.

We have now incorporated this into the Discussion section on Pages 8-9, Lines 700-771. However, we caution that these findings should be interpreted with care given the smaller sample size, and validation in larger, independent studies is warranted.

Page 5, Lines 174-177:

Notably, the Asian population exhibited the highest average model performance, with a mean R^2 of 0.18 (Figure 1b). Kruskal–Wallis test revealed a significant difference in R^2 values across the four groups ($\chi^2 = 98.4$, $df = 3$, $P < 2.2 \times 10^{-16}$). Post hoc Dunn’s tests with Bonferroni correction showed that R^2 values were significantly higher in Asians compared to Africans ($P < 2.6 \times 10^{-5}$), Europeans ($P < 7.8 \times 10^{-22}$), and Hispanic/Latino populations ($P < 1.3 \times 10^{-5}$).

Pages 8-9, Lines 705-778:

Interestingly, the models for the Asian population exhibited the highest average cross-validated R^2 value of 0.18, despite this being the smallest subgroup within the MESA cohort. This suggests that, in this population, certain proteins are more strongly or consistently regulated by genetic variants, leading to improved predictive performance at specific loci. This hypothesis is further supported by our Kruskal–Wallis test, which revealed significant differences in heritability (H^2) values across the four groups ($\chi^2 = 156.89$, $df = 3$, $P < 2.20 \times 10^{-16}$). In post hoc Dunn’s tests, H^2 values were significantly higher in Asians compared to Africans ($P = 0.02$), Europeans ($P = 4.38 \times 10^{-29}$), and Hispanic/Latino participants ($P = 2.69 \times 10^{-3}$). While these results are encouraging, they should be interpreted with caution given the limited sample size. Future studies leveraging larger and more diverse Asian cohorts will be essential to validate and further investigate these observations.

Page 14, Lines 1041-1044:

We compared R^2 and H^2 across African, European, Asian, and Hispanic/Latino populations using the Kruskal–Wallis test, given the non-normal distribution of R^2 and H^2 values. When significant, post hoc pairwise comparisons were performed using Dunn’s test with Bonferroni correction. Analyses were conducted in R (version 4.1.2) using the ‘FSA’ package.

Reviewer #3

Comment 1: *Incomplete sentence on page 4 line 105.*

Response-1:

We thank the reviewer for catching this oversight. We have now revised it for clarity and completeness in the revised manuscript. We appreciate your careful reading and valuable feedback.

Page 4, Lines 124-127:

PCa incidence rates vary significantly by regions, with the lowest rates observed in South Central Asia (6.3 age-standardized rate per 100,000), and markedly higher rates in Northern Europe (82.8 age-standardized rate per 100,000).

Comment 2: *Why only 6,999 out of 7,367 plasma proteins were tested (page 6 line 141)?*

Response-2:

We thank the reviewer for this question. We have clarified this in the Methods section on Page 13, Lines 999-1005. In this study, proteomic profiling was performed using the aptamer-based SomaScan assay, which initially quantified 7,289 human proteins. We excluded 10 SOMAmers targeting proteins whose encoding genes lacked positional information in the BioMart database. Additionally, 280 SOMAmers targeting proteins encoded on sex chromosomes were removed to focus our analysis on plasma proteins or protein complexes encoded by autosomal genes. After these exclusions, a total of 6,999 proteins were retained for downstream analysis.

Page 13, Lines 999-1005:

Proteomic Data Processing

Proteomic profiling was conducted using the aptamer-based SomaScan assay, which quantified 7,289 human proteins. We excluded 10 SOMAmers whose target protein-encoding genes lacked positional information in the BioMart ⁷⁵ database. An additional 280 SOMAmers targeting proteins encoded on sex chromosomes were removed to focus the analysis on plasma proteins or protein complexes encoded by autosomal genes. After these exclusions, 6,999 proteins were retained for downstream analysis.

Comment 3: *What was the rationale of using a FDR of 0.05 for cis-pQTLs while trans-pQTLs were identified at $P < 5 \times 10^{-9}$?*

Response-3:

We thank the reviewer for this important question. The use of different significance thresholds for cis- and trans-pQTLs reflects standard practice in proteogenomic studies and is based on the

differing multiple testing burdens and prior probabilities of association. For *cis*-pQTLs, we limited the analysis to variants within ± 500 kb of the encoding gene, significantly reducing the number of tests; therefore, a FDR of 0.05 was applied to balance sensitivity and specificity. *Trans*-pQTLs were tested genome-wide, requiring a more stringent threshold to control for the substantially larger numbers of comparisons and minimize false positives (Liu et al., 2024). Compared with the typical threshold of $P < 5 \times 10^{-8}$ in GWAS, we used $P < 5 \times 10^{-9}$ as we focused on multiple proteins. This represents a tradeoff between sensitivity and specificity. This rationale has now been added to the Methods section on Page 14, Lines 1021-1022 for clarity. We appreciate your thoughtful suggestion.

Page 14, Lines 1021-1022:

These significance thresholds were chosen to maximize the inclusion of potentially informative SNPs while minimizing excess noise⁸³.

Comment 4: *How do you determine the cutoff of performance $R^2 \geq 0.01$ for external validation?*

Response-4:

We thank the reviewer for this important question. The cutoff of cross-validated $R^2 \geq 0.01$ for model inclusion in external validation was chosen based on established precedents in proteome-wide association studies and transcriptome-wide association studies (TWAS) (e.g., Gamazon et al., 2015; Zhong et al., 2023; Liu et al., 2024; He et al., 2025). This threshold balances retaining models with meaningful predictive power while excluding poorly performing models that are unlikely to produce reliable imputation results. An R^2 of 0.01 corresponds to a 10% correlation between genetically predicted expression vs directly measured expression, which, has been proposed in the original PrediXcan method and multiple prior PWAS and/or TWAS studies (e.g., Wu et al., 2020; Gamazon et al., 2015; Zhong et al., 2023; Liu et al., 2024; Zhu et al., 2024; He et al., 2025) to yield useful and replicable association signals when applied in large GWAS datasets. We have now clarified this rationale in the Methods section on Page 14, Lines 1031-1034 and Page 15, Lines 1123-1124, citing relevant literature to support the choice.

Page 14, Lines 1031-1034:

Only models with a cross-validation $R^2 > 0.01$ (indicating they explain more than 1% of the variance, corresponding to a minimum ~10% correlation between predicted and measured protein levels) were included in subsequent association analysis. This threshold is commonly applied in similar studies^{9,83,88-91}.

Page 15, Lines 1123-1124:

Models with a performance R^2 value ≥ 0.01 (at least ~10% correlation between predicted and measured protein levels) were considered to have successfully passed external validation^{18,83}.

Comment 5: *Is it a typo on page 19 line 442?*

Response-5:

Thank you for pointing this out. We have reviewed the sentence on page 19, line 442 (now on page 15, line 1139), and confirmed that it was a typographical error. The character “0” should have been a closing parenthesis “)”. This has been corrected in the revised manuscript. We appreciate your careful attention to detail.

Page 15, Lines 1137-1139:

Evidence of heterogeneity of the associations across racial/ethnic groups was assessed using the I^2 statistic in METAL, with high heterogeneity defined as $I^2 > 75$ and a heterogeneity P -value ($HetPVal$) < 0.05 ⁹⁴.

Comment 6: In Figure 3, it would be helpful to indicate whether the N was for number of subjects or for number of proteins.

Response-6:

Thank you for this helpful suggestion. We have updated Figure 3 to clarify whether “N” represents the number of proteins, subjects, or models, as appropriate. We appreciate your feedback in helping to improve the clarity of the presentation.

Figure 3. The overall design of this study.

Comment 7: *The authors mentioned that the models with a cross-validation $R^2 > 0.01$ were used for further association analysis. It was not clear how the cross-validation was conducted within each population. Also, the cross-validation $R^2 > 0.01$ appears to be low. The same cutoff was used for external validation for European population. Why not focus on models with a higher R^2 to focus on proteins with better genetic predictions which could also improve the power of this study?*

Response-7:

Thank you for this important and insightful comment. We appreciate the opportunity to clarify our cross-validation approach and rationale for the R^2 cutoff. For each population-specific protein prediction model, we performed five-fold cross-validation within the MESA proteomics training dataset by partitioning the data into five subsets, training the model on four folds, and evaluating predictive performance on the held-out fold, iterating so that each fold served as the test set once. The resulting cross-validated R^2 represents the average proportion of variance explained across all folds. We applied a threshold of $R^2 \geq 0.01$, a commonly used minimal cutoff in proteome- and transcriptome-wide association studies (e.g., Wu et al., 2020; Gamazon et al., 2015; Zhong et al., 2023; Liu et al., 2024; Zhu et al., 2024; He et al., 2025), to balance retaining broad protein coverage with ensuring sufficient predictive performance. Although higher thresholds (e.g., 0.05 or 0.1) would enrich for more robust models, they would also markedly reduce the number of proteins available for testing, potentially limiting discovery. The same $R^2 \geq 0.01$ threshold was applied in external validation within the European population to ensure consistency and comparability across groups, while our downstream analyses incorporated multiple-testing correction and biological prioritization to identify robust associations.

We have now clarified these points in the Methods sections on Page 14, Lines 1031-1040 and Page 15, Lines 1123-1125.

Page 14, Lines 1031-1040:

Only models with a cross-validation $R^2 > 0.01$ (indicating they explain more than 1% of the variance, corresponding to a minimum ~10% correlation between predicted and measured protein levels) were included in subsequent association analysis. This threshold is commonly applied in similar studies^{9,83,88-91}. Cross-validation was performed using a five-fold scheme, where the dataset was randomly divided into five equal parts. In each fold, models were trained on 80% of the data and tested on the remaining 20%, rotating such that each subset served as a test set once. The final cross-validation performance was calculated by regressing observed protein levels against the predicted values aggregated from all test folds. This adjusted R^2 accounts for model complexity and sample size, providing a conservative measure of the variance explained by the genetic predictors.

Page 15, Lines 1123-1125:

Models with a performance R^2 value ≥ 0.01 (at least ~10% correlation between predicted and measured protein levels) were considered to have successfully passed external validation^{18,83}.

RE: NCOMMS-25-32305-B

Title: Proteome-wide association study of prostate cancer risk across populations

Reviewer #1 (Remarks to the Author):

The authors have fully addressed all my comments. This reviewer appreciated their major efforts to compare with other types of evidence (PCa GWAS, TWAS, etc.) and further interpretation. The new results solidified their results, making their conclusion more reliable. Therefore, the revised manuscript has had many improvements for high quality. I do not have any further comments except for one minor suggestion below, which I believe the authors can address well in the final version (without my further review).

Comment 1. I appreciated the authors provided a detailed Suppl Table 1. How are these rows ordered? In addition, Column “Target” is for protein and Column “EntrezGeneSymbol” is for gene symbols. Many protein abbreviations in Target column are based on protein symbols. The authors may double check to see some of abbreviations can be better by using protein names, or provide the source of such protein abbreviations in the table footnote.

Response-1:

We appreciate the reviewer’s helpful suggestion regarding Supplementary Table 1. The rows in the table are ordered first by population information and then by protein ID. The abbreviations in the "Target" column are based on the protein annotations provided by SomaLogic, the platform used for protein quantification in the MESA study. To improve clarity, we have now included a footnote in Supplementary Table 1 clarifying that the protein abbreviations are derived from SomaLogic’s annotation file.